# Which factors should be included in triage? An online survey of the attitudes of the UK general public to pandemic triage dilemmas

Dominic Wilkinson  ,[1,2,3,4] Hazem Zohny,[1] Andreas Kappes,[5] Walter Sinnott-Armstrong,[6] Julian Savulescu[1,3,4,7]

For numbered affiliations see end of article.

**Correspondence to**
Dr Dominic Wilkinson;
dominic.wilkinson@philosophy.ox.ac.uk

## ABSTRACT

**Objective** As cases of COVID-19 infections surge, concerns have renewed about intensive care units (ICUs) being overwhelmed and the need for specific triage protocols over winter. This study aimed to help inform triage guidance by exploring the views of lay people about factors to include in triage decisions.

**Design, setting and participants** Online survey between 29th of May and 22nd of June 2020 based on hypothetical triage dilemmas. Participants recruited from existing market research panels, representative of the UK general population. Scenarios were presented in which a single ventilator is available, and two patients require ICU admission and ventilation. Patients differed in one of: chance of survival, life expectancy, age, expected length of treatment, disability and degree of frailty. Respondents were given the option of choosing one patient to treat or tossing a coin to decide.

**Results** Seven hundred and sixty-three participated. A majority of respondents prioritised patients who would have a higher chance of survival (72%–93%), longer life expectancy (78%–83%), required shorter duration of treatment (88%–94%), were younger (71%–79%) or had a lesser degree of frailty (60%–69%, all p<0.001). Where there was a small difference between two patients, a larger proportion elected to toss a coin to decide which patient to treat. A majority (58%–86%) were prepared to withdraw treatment from a patient in intensive care who had a lower chance of survival than another patient currently presenting with COVID-19. Respondents also indicated a willingness to give higher priority to healthcare workers and to patients with young children.

**Conclusion** Members of the UK general public potentially support a broadly utilitarian approach to ICU triage in the face of overwhelming need. Survey respondents endorsed the relevance of patient factors currently included in triage guidance, but also factors not currently included. They supported the permissibility of reallocating treatment in a pandemic.

## Strengths and limitations of this study

► First UK survey to investigate public attitudes to pandemic triage dilemmas.
► Large survey, representative of the UK general population.
► Enables comparison of ethical arguments and existing guidance with the views of the public.
► Identifies relevance of specific patient factors in concrete forced choice dilemmas: may be helpful in development or revision of triage policies.
► Survey findings do not allow assessment of relative weight of different factors or how they might interact.

patients presenting with severe COVID-19.[1 2] In early March in Northern Italy, one of the worst affected regions of Europe, hospitals and intensive care units (ICUs) were overwhelmed.[3 4] Pandemic modelling in the UK suggested that high rates of infection with the virus in the UK would exceed the availability of intensive care.[5]

Faced with such concern, health systems around the world prepared guidance for intensive care triage.[6–8] The aim was to help health professionals make extremely difficult and potentially distressing decisions[9] about which patients to admit to ICU and treat with mechanical ventilation. There was active ethical and political debate about which patient factors should or should not be included in triage.

Broadly, prioritisation decisions relating to scarce treatments can be based on three different ethical approaches. In the context of allocating health resources, utilitarianism seeks to maximise total population health, for example by prioritising patients with the best prognosis. Egalitarianism highlights equal treatment for equal need and underpins the UK National Health Service (NHS).[10 11]

## BACKGROUND

In the first phase of the coronavirus pandemic, there was widespread concern that there would be insufficient intensive care beds and mechanical ventilators to treat the number of

Prioritarianism gives priority to the worst off; this is sometimes interpreted as giving priority to those with greatest medical need or who are medically most vulnerable.[12–14] (These descriptions are necessarily somewhat simplified and constrained to allocation of health resources. Applied as guiding principles for a society or the entire human population, they may have different implications. For instance, if prioritising those with the best prognosis may, in a particular political or cultural context, lead to worse overall well-being, utilitarianism may favour a more egalitarian or prioritarian approach. Prioritarianism might imply priority for patients who are worse off in other ways (for example having experienced social or economic disadvantage). Non-utilitarian theories may also support maximising numbers of lives or life-years saved.[15])

For COVID-19, a number of different patient factors might be relevant for ICU triage. Some factors influence the number of people who would benefit.[16] Saving as many lives as possible is arguably a fundamental ethical principle for any triage framework, and supported by overlapping consensus of different ethical theories.[15] Prioritisation of patients more likely to survive, or those likely to need shorter duration of treatment, would lead to more survivors overall. Other patient factors are relevant to the *magnitude* of benefit. For example, prioritising patients with a longer life expectancy or less pre-existing disability would not save more lives, but would result in more quality-adjusted life-years.[17] However, inclusion of such factors might be vulnerable to bias and raise concerns about discrimination.[15] Other patient factors could be relevant in more than one way. Patient age appears to be a risk factor for mortality in COVID-19, but is also relevant to life expectancy. Separately, some have argued that younger patients deserve to be prioritised as they have not had a chance to live a complete life.[11] Clinical frailty in patients requiring intensive care admission has been widely researched as a potential prognostic factor and triage tool. It is potentially relevant to patient survival, length of life and quality of life.[18]

In some parts of the world, pre-existing triage guidelines for an influenza pandemic had been informed by prior community consultation. For example, a series of community engagement forums in Maryland over 2012–2014 identified support for prioritisation to save the most lives and life-years, but also evinced a concern about reallocating (withdrawing) treatment once commenced.[19 20] That evidence was adapted and incorporated into statewide guidance for COVID-19 in Maryland and elsewhere.[21 22]

In the UK, to our knowledge, there have been no prior public surveys. The National Institute for Health and Care Excellence (NICE) published a rapid clinical guideline on critical care for adults in the context of COVID-19 on 20th of March 2020.[8] The only specific factor mentioned was clinical frailty. A draft UK national pandemic allocation guideline, developed in late March, proposed a scoring system incorporating age, frailty and comorbidities.[23] This guideline was apparently rejected

by the UK health officials,[24] and no official NHS guidance was produced.

In the first phase of the pandemic, intensive care resources were not overwhelmed in the UK, and explicit rationing was not required.[25] However, there remain concerns about a further surge of cases in the coming months or in future pandemics. Given this possibility and the potential value of gauging community views at the most relevant time, we aimed to explore the view of lay people about resource allocation decisions. To identify which patient factors are thought by the public to be relevant, we used a series of hypothetical rationing dilemmas based on our prior work evaluating resource allocation in neonatal intensive care.[26] While the general public's views do not fully resolve questions about which approaches should be adopted, they are relevant to the goals of democratic legitimacy and may play an important role in achieving reflective equilibrium.[27 28]

## METHODS

We conducted a survey of the UK residents from 29th of May to 22nd of June 2020. Participants were recruited from an established online platform (https://www.qualtrics.com). They were sampled to be representatives of the UK general population for gender, age, household income, education and employment. Participants were recruited from existing large market research panels and were remunerated at a rate of £8/hour. Attention checks and speed checks were used to identify respondents not paying sufficient attention to question details. Seven hundred and sixty-three participated. Their mean age was 44±15 years, (range 18–86). Fifty-four per cent were women (for full demographic characteristics, see online supplemental table 1). A sample size of 500 or higher was estimated to provide power of 0.95 to detect even small differences in preferences within subjects between the different scenarios.[29]

The survey was designed to assess participants' views about incorporating patient factors into prioritisation decisions, if there are insufficient ventilators to treat all patients who require them during the COVID-19 pandemic. It was adapted from a previous survey on rationing in neonatal intensive care.[26] Survey questions were modified to relate to adult patients with COVID-19 (full survey text: https://osf.io/gta3k/).

The survey tested which characteristics would lead to priority in scenarios where two patients require treatment for COVID-19, but where only one ventilator is available.

Participants were given the following preamble:

We would like you to imagine you are a doctor in the intensive care unit (ICU) of a hospital in the UK… As you are probably aware, one of the challenges of the current coronavirus pandemic is that many patients may become unwell at the same time… You will be required to make decisions about whether or not to provide treatment in the ICU. In the cases we are

**A**

There are two patients who are seriously ill with coronavirus who have come to hospital around the same time. Both require ICU admission. **The ICU is almost full. There is only one ventilator available, so only one patient can be treated. The other patient is likely to die.**

You have assessed them and estimate that they both have **the same life expectancy** and **a similar quality of life in the future** if they survive.

Patient A has an **80%** chance of survival with treatment.

Patient B has a **10%** chance of survival with treatment.

Do you:

○ Treat patient A

○ Treat patient B

○ Toss a coin to decide

**B**

**The ICU is almost full. There is only one ventilator available, so only one patient can be treated. The other patient is likely to die.**

There are two patients who are unwell with coronavirus and both require ICU admission. They have an **equal chance of survival**. Based on their underlying health you estimate that they are **both expected to live for 15 years** if they recover from this illness.

Patient A is **71 years old**.

Patient B is **55 years old**.

Do you:

○ Treat patient A

○ Treat patient B

○ Toss a coin to decide

**C**

There are two patients who are unwell with coronavirus and both require ICU admission. They both have **an equal chance of survival and are 50 years old**.

Patient C has **no pre-existing disability**.

Patient D has **profound and multiple learning disabilities**. This patient has difficulty seeing, hearing, speaking and moving and are dependent on others for self-care. This patient lives in a nursing home and has a mental age of less than 3 years.

Do you:

○ Treat patient C

○ Treat patient D

○ Toss a coin to decide

**Figure 1** Hypothetical pandemic triage dilemmas. (A) Example question with varying chance of survival; (B) example question with varying age; (C) example question with varying degrees of disability. ICU, intensive care unit.

discussing in this survey, if they are not treated, the patients are likely to die. In all of the situations discussed in the survey, the patients have indicated that they would like life-saving treatment to be provided.

They were presented with a series of 38 allocation/withholding scenarios in which only a single ventilator is available, and two patients require ICU admission and ventilation. The patients differed in one of six key variables: chance of survival, life expectancy, age, expected length of treatment, disability and degree of frailty (see figure 1). For a given scenario, variables were assumed to be known or predicted accurately and (unless specified) other characteristics were the same between patients. Degrees of disability and frailty were described in terms of

severity and impact on daily life. (The term 'learning disability' was used in the survey as this is a common accepted term to refer to intellectual disability in the UK.[30] An approximate equivalent 'mental age' was included alongside the description of function to aid participants' understanding, though this no longer features in official classification of intellectual/learning disability.[31] Degrees of frailty were described in accordance with the Clinical Frailty Score.[32]) For scenarios relating to duration of treatment, participants were informed that if treatment was allocated to the patient expected to need a shorter period of intensive care, one or more additional subsequent patients would be able to be treated. Participants were instructed that patients were relevantly similar apart

from the key variable. They were asked to choose one of the patients to treat or to toss a coin to decide. Scenario blocks were presented in random order, and the order of scenarios within each block was also randomised. Scenarios varied in whether the first or second patient had the higher value of the relevant variable.

A set of three allocation/withdrawal scenarios were presented where participants were given a choice between continuing (or withdrawing) treatment for a patient who had been in intensive care for 2 weeks and was not improving, and commencing (or withholding) treatment for a patient with a higher chance of survival who arrived in the hospital today. (The survival chance differences were identical to those used in allocation/withholding scenarios.)

A follow-up scenario for questions relating to chance of survival included a 'veil of ignorance', designed to elicit impartial judgements.[26] Participants were asked which allocation policy they would choose if they knew that a family member would become unwell and need a ventilator later in the year, but without knowing whether their relative would have a higher or lower chance of survival.

Follow-up questions relating to age and frailty sought to establish whether participants' views altered if the variable affected either survival chance or longevity. For example, participants were told that for two patients of different ages, their survival chances had been estimated based on their age.

Finally, respondents were asked in separate scenarios to choose between patients who had different numbers of dependents, and between patients, one of whom was a healthcare worker, working in a hospital, the other of whom was a non-key worker, working from home. A control scenario asked participants to choose to allocate treatment between patients of different racial backgrounds.

For analysis, selection of the patient with the higher (better) level of the relevant variable was coded as a 'Better prognosis' response. Choosing to toss a coin was coded as 'Equal chance'. Choosing the lower (worse) level of the relevant variable was coded as a 'Worse prognosis' response. For a scenario where patients had different types of disability, these were coded for the type of disability.

### Statistical analysis

To test whether distributions between scenarios differed, we computed McNemar-Bowker tests (paired/matched $X^2$ tests), which allow for within-subject comparisons of distributions. Furthermore, to control for the repeated testing of the same hypothesis on different scenarios, we corrected for multiple comparisons using Bonferroni correction. We report below the corrected significance values.

As an exploratory analysis, we examined the relationship of demographic variables to an index indicating how often participants decided to choose the better prognosis treatment option on each of the 33 dilemmas. We compared responses by gender, education and household income.

### Patient and public involvement

No patient involved.

## RESULTS

Participants were prescreened for demographic characteristics and excluded (prior to participation) after quotas for demographic subgroups were met. Three hundred and seventeen participants were excluded for failing one of the attention checks. Thirty-four participants were excluded for completing the survey in less than half the median completion time. Seven hundred and sixty-three respondents completed the survey and were included for analysis.

For each scenario, responses differed significantly from chance, indicating clear preferences (p values all p<0.001).

### Survival

A large majority of respondents elected to allocate treatment to a patient with a higher chance of survival in three of four scenarios (figure 2A). As the difference between the patients decreased, a larger proportion of participants chose the equal-chance option. For patients with a very small difference in predicted survival (49% vs 51%), approximately half of participants chose to toss a coin.

When participants were asked a 'veil of ignorance' variant of this question, 90% chose the patient with a better prognosis.

### Ventilator withdrawal

In three out of four scenarios, a clear majority of respondents elected to remove ventilator treatment if that allowed a patient with higher survival chance to receive treatment (figure 2B). When the difference in survival was greater, a larger proportion of participants chose the better prognosis option. However, when the difference in survival chance was small (49% vs 51%), 34% prioritised the patient with better chances and 37% elected to toss a coin.

In all scenarios, a somewhat smaller proportion of participants chose the better prognosis option when withdrawing than in the equivalent withholding scenarios

### Life expectancy

A clear majority of respondents elected to allocate treatment to patients with greater life expectancy in three of four scenarios (figure 3). For patients with a very small difference in life expectancy (15 vs 14 years), 55% chose to toss a coin. Equal chance (coin toss) responses increased with smaller difference in life expectancy.

### Age

A large majority of respondents elected to allocate treatment to a younger patient rather than an older patient in three of four scenarios where life expectancy and survival

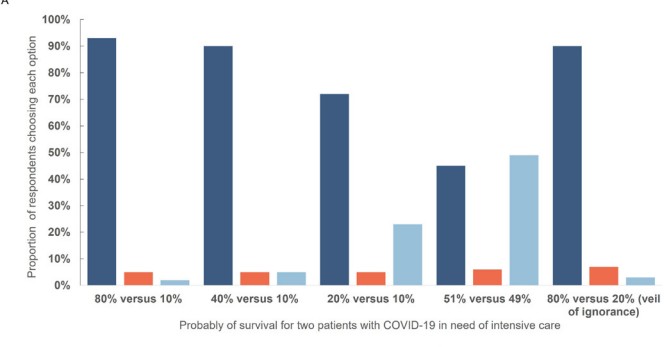

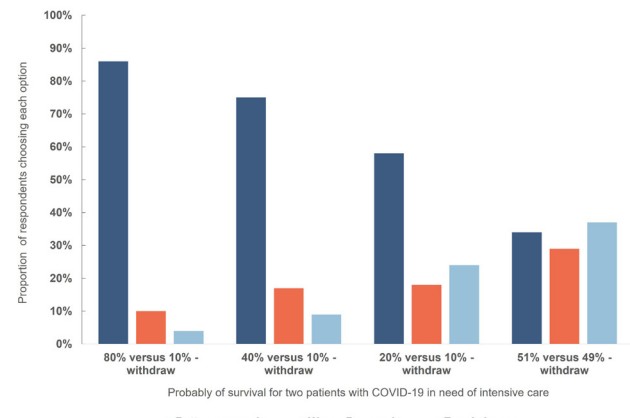

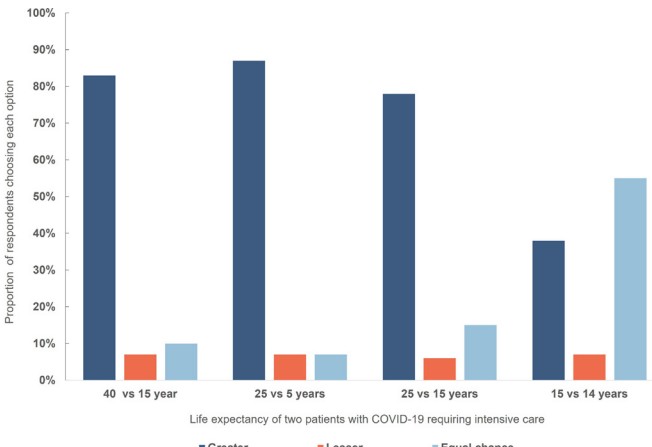

**Figure 2** (A) Respondent choices in a triage dilemma involving withholding treatment from one of two patients with different survival chances. There was a statistically different distribution in responses when comparing the 80% vs 10% with the 40% vs 10% chances of survival scenario ($X^2$ (3, N=763)=19.793, p<0.001), 80% vs 10% with the 20% vs 10% scenario ($X^2$ (3, N=763)=165.077, p<0.001), 80% vs 10% with the 51% vs 49% chances of survival scenario ($X^2$ (3, N=763)=371.54, p<0.001). Similarly, we found statistically different distribution in responses when comparing the 40% vs 10% with the 20% vs 10% scenario ($X^2$ (3, N=763)=143.00, p<0.001); 51% vs 49% scenario ($X^2$ (3, N=763)=351.298, p<0.001) and 20% vs 10% with the 51% vs 49% ($X^2$ (3, N=763)=198.278, p<0.001). (B) Respondent choices in a triage dilemma involving patients with different survival chances where the patient with worse prognosis was already receiving treatment in intensive care. There was a statistically different distribution in responses when comparing these scenarios with equivalent withholding versions: 80/10: $X^2$ (3, N=763)=27.766, p<0.001; 2: 40/10: $X^2$ (3, N=763)=87.105, p<0.001; 3: 20/10: $X^2$ (3, N=763)=81.977, p<0.001; 51/49 $X^2$ (3, N=763)=180.061, p<0.001.

chance were said to be equal (figure 4A). For patients with a very small difference in age (72 vs 71 years), 65% of respondents chose to toss a coin.

More participants chose the younger patient and fewer the equal chance option when there was a larger difference in patient age.

When participants were given additional versions of the cases in which the patient's age was linked with survival chances, a higher proportion of respondents elected to allocate treatment to the younger (and more likely to survive) patient (figure 4B).

**Figure 3** Respondent choices in a triage dilemma involving withholding treatment from one of two patients with different life expectancy. There was a statistically different distribution in responses: 1: 25/5 years vs 40/15 years: $X^2$ (3, N=763)=23.89, p<0.001; 2: 25/5 vs 25/15: $X^2$ (3, N=763)=62.562, p<0.001; 3: 25/5 vs 15/14: $X^2$ (3, N=763)=305.042, p<0.001.

### Length of treatment
A very large majority of respondents elected to allocate treatment to patients requiring shorter periods of treatment (with the expectation that this would enable more patients to be treated) (figure 5).

More participants chose the better prognosis patient when there was a large difference in expected duration of treatment.

### Disability
In three out of four scenarios involving patients with different degrees of pre-existing disability, a similar proportion of respondents elected to treat a patient with lesser or no disability as elected to toss a coin (figure 6). A majority of respondents (74%) elected to allocate treatment to a non-disabled patient in preference to a patient with profound learning disability. A minority of respondents (11%–19% in the different scenarios) elected to treat the patient with greater disability.

### Frailty
In scenarios where chance of survival and life expectancy were said to be the same, a majority of respondents elected to allocate treatment to less frail patients compared with more frail ones (online supplemental figure 1A). In the scenario with mild versus no frailty, 49% chose the patient with no frailty. Similar proportions of participants chose the better prognosis option when deciding between patients with severe and moderate frailty compared with patients with severe and mild frailty.

When participants were given additional versions of cases in which the patient's degree of frailty was associated with either reduced survival chance or reduced life expectancy, a larger proportion of respondents elected to allocate treatment to the less frail patient (online supplemental figure 1B). When participants were given

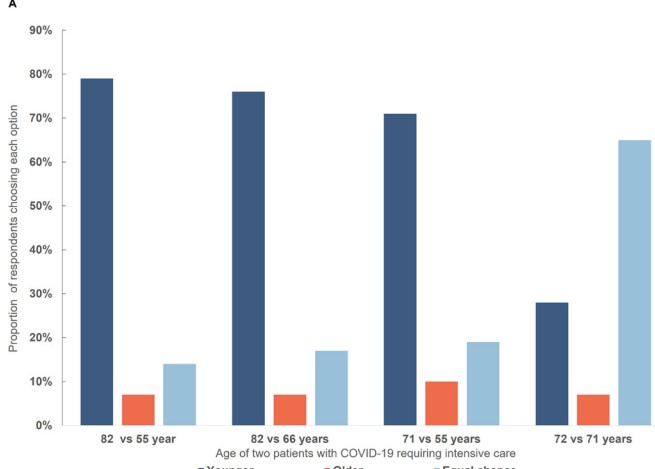

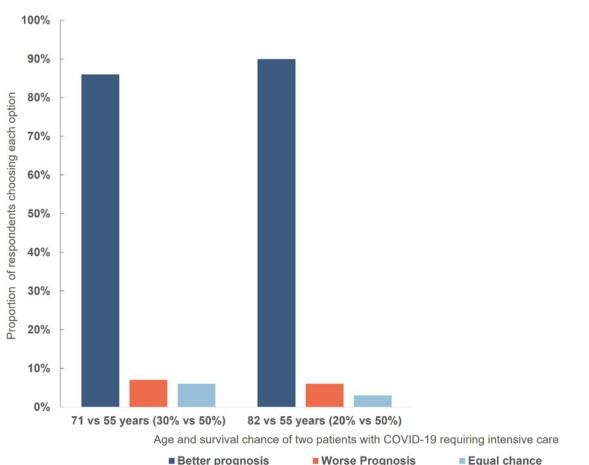

**Figure 4** (A) Respondent choices in a triage dilemma involving withholding treatment from one of two patients with different age but identical survival chance/life expectancy. There was a statistically different distribution in responses: 1: 82/55 vs 82/66: $X^2$ (3, N=763)=19.455, p<0.001; 2: 82/55 vs 71/55: $X^2$ (3, N=763)=47.608, p<0.001; 3: 82/66 vs 71/55: $X^2$ (3, N=763)=25.64, p<0.001. Responses on all scenarios differed significantly when compared with the 72/71 scenario, $X^2$>1061.42, p<0.001. (B) Respondent choices in a triage dilemma involving withholding treatment from one of two patients with different age where the older patient had a lower survival chance. There was a significant difference in distribution of responses compared with equivalent scenarios where survival chance was said to be identical: 71/55: −$X^2$ (3, N=763)=95.65, p<0.001; 82/55: $X^2$ (3, N=763)=77.219, p<0.001.

a case requiring a choice between a younger but more frail patient, and an older but less frail patient, a larger proportion elected to treat the less frail, older patient (44% vs 31%).

## Other variables

In a question requiring a choice between two patients of different racial backgrounds, 66% elected to toss a coin, 18% elected to treat a white British patient, while 16% chose to treat a patient of Black Caribbean background.

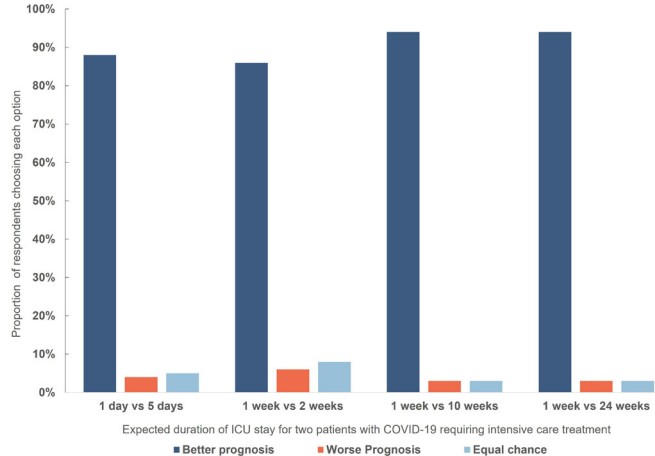

**Figure 5** Respondent choices in a triage dilemma involving withholding treatment from one of two patients with different expected duration of treatment. There was a significant difference in the distribution of answers between scenarios: 1: 24 weeks/1 week vs 1 week/2 weeks: $X^2$ (3, N=763)=42.66, p<0.001; 2: 24 weeks/1 week vs 5 days/1 day: $X^2$ (3, N=763)=30.047, p<0.001; 3: 24 weeks/1 week vs 10 weeks/1 week: $X^2$ (3, N=763)=1.085, p=0.99; 4: 2 weeks/1 week vs 5 days/1 day: $X^2$ (3, N=763)=2.798, p=0.99. ICU, intensive care unit.

Asked to choose between two patients, one of whom was a healthcare worker, 63% elected to treat the healthcare worker, while 33% chose to toss a coin.

Finally, in a scenario of two patients with similar characteristics, one of whom had three young children, while the other had no dependents, 80% elected to treat the patient with young children and 18% chose to toss a coin.

## Relationship between demographic variables and responses

Gender and household income did not affect participants' tendency to choose the patient with better prognosis (gender t(760)=1.291, p=0.197; income F(1750)=2.43,

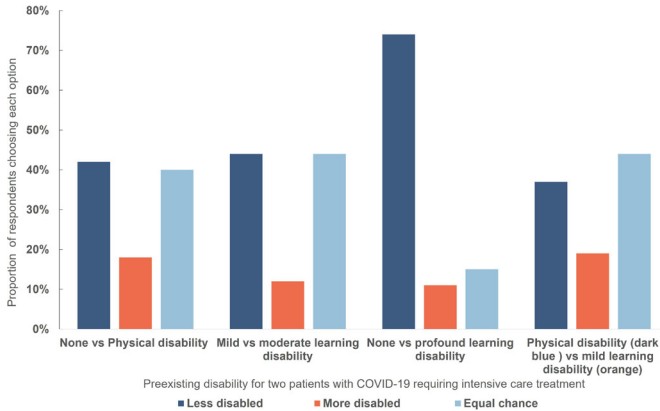

**Figure 6** Respondent choices in a triage dilemma involving withholding treatment from one of two patients with different degrees of pre-existing disability. There was a significant difference in the distribution of answers between scenarios: 1: profound learning disability (LD)/none vs moderate/mild $X^2$ (2, N=763)=536.177, p<0.0001; 2: profound LD/none vs physical/none $X^2$ (2, N=763)=464.653, p<0.0001.

p=0.088). There was a significant difference for education; those with higher reported levels of education choosing less often the better prognosis patient than those who reported lower levels of education t(760)=3.672, p<0.001.

## DISCUSSION

This survey, conducted at the end of the first wave of the COVID-19 pandemic in the UK, is the first to assess the views of the general UK public about patient factors that are relevant for triage. In this large survey, designed to include participants representative of the UK general population, a majority elected to prioritise patients with better prognosis in a way that would maximise health-care benefit (in line with a utilitarian approach to triage). Presented with a set of hypothetical COVID-19 triage dilemmas, respondents prioritised patients who would have a higher chance of survival, longer duration of survival, shorter duration of treatment, lower age or lesser degree of frailty. Where there was a small difference between two patients, a larger proportion elected to toss a coin to decide which patient to treat. Respondents were more egalitarian in scenarios relating to patients with pre-existing disability. A majority were prepared to withdraw treatment from a patient in intensive care who had a lower chance of survival than another patient currently presenting with COVID-19. More participants were prepared to withhold than withdraw life-saving treatment in equivalent cases. Respondents also indicated a willingness to give higher priority to healthcare workers and to patients with young children.

### Previous surveys

The overall results of this survey are very similar to our previous survey focused on neonatal intensive care.[26] In that survey, more than three-quarters of US-based respondents elected to treat a newborn infant with better predicted outcome, but more elected to toss a coin when the differences between patients became small. In the current survey, respondents appeared more inclined to toss a coin when choosing between patients with different types or degrees of disabilities (~40% of respondents in three scenarios in this survey, compared with ~20% in the previous survey). This might reflect public concern about the ethical problems in assessing quality of life,[15] or a desire to avoid discrimination.

Prior studies of community attitudes to pandemic triage have often indicated support for prioritisation that would aim to save the most lives and life-years (table 1). For example, community engagement forums in Maryland in 2012–2014 identified the importance of saving the most lives and the most life-years.[20] These same principles were recently endorsed by a similar deliberative process in Texas.[33]

Our survey findings were somewhat different from another general public survey conducted during the COVID-19 pandemic. Buckwalter and Peterson conducted an online survey with US respondents. Participants indicated support for triage policies that aimed to save the most lives ('utilitarian' policy) or treat the sickest patients (labelled 'prioritising the worst off'), but disagreed with policies that treated patients in order of arrival ('egalitarian') or prioritised based on social importance.[34] However, the results of Buckwalter and Peterson's study are hard to compare with our own. Participants in that study were asked to endorse general policies, but not presented with specific cases of competing patients. The policy descriptions might not have sufficiently distinguished ordinary judgements related to maximising benefits versus prioritising patients who are the worst off since some participants who chose 'prioritising the worst off' policy might have intuitively believed that sicker patients would benefit most from treatment (though in fact sicker patients may have a lower chance of survival).

Our survey found that participants endorsed treating younger rather than older patients, if forced to choose. They prioritised a younger patient even if told that both patients had identical survival chances and duration of expected survival. They were even more likely to prioritise the younger patient in a situation where older age was linked with lower chance of survival. This finding diverges from a study on public attitudes in Germany, which did not find support for age as a criterion relevant to prioritising healthcare.[35] It is, however, broadly consistent with Buckwalter and Peterson's survey which found agreement with a utilitarian triage policy even if it disadvantaged older patients.[34] It is also consistent with community studies that mentioned maximising numbers of life-years saved,[20 33] as well as recent study using veil-of-ignorance reasoning in COVID-19 ventilator dilemmas.[36] While this latter study found older participants do not initially favour prioritising the young, they do *after* imagining that they did not know if they would be the younger or older patient requiring a ventilator to survive.

Our participants demonstrated a nuanced inclusion of age in decisions. In a scenario involving a choice between a more frail younger patient (aged 66) and a less frail older patient (aged 82), a higher proportion chose the older patient. In our survey, we also asked about the use of frailty in triage decisions. To our knowledge no previous studies have specifically gauged attitudes to frailty in such decisions. A majority of our respondents prioritised treatment for a less frail patient, and this increased when frailty was associated with survival chance or longevity.

Our survey respondents indicated a willingness to prioritise scarce treatment for healthcare workers and for patients with dependents (young children). These factors are not commonly included in triage guidelines, but both have been mentioned in community deliberation relating to pandemic planning.[20 33] Residents of Central Texas placed particular importance on the importance of family, and some mentioned that priority might be given to those with family who depend on them.[33]

One important difference between the current survey and other studies is in the support for withdrawal of treatment to allow re-allocation to another patient with

**Table 1** Previous studies of community attitudes to pandemic and/or disaster triage

| Study | Description | Key findings relevant to triage |
|---|---|---|
| Ritvo et al[45] | Canadian telephone survey administered in 2009 to a random sample of 500 Canadians to obtain opinions on key ethical issues in pandemic preparedness planning | Mortality reduction, with priority to children, healthcare workers infected while serving patients, the sickest patients and adults with dependents |
| Vawter et al[46] | Minnesota public engagement initiative taking place in 2009 | Keyworkers should not be prioritised. Refrain from rationing ventilators based on differences in socioeconomic status, quality of life, life expectancy or first-come, first-served |
| Li-Vollmer[47] | Washington state public engagement meetings conducted from 2008 to 2009 | Prioritisation of medical services should aim to save the greatest number of people, factoring in survivability of those treated, even if standards of care must be lowered. Priority also to first responders and healthcare workers, with children and pregnant women given some priority when all other factors are equal. Overwhelming rejection of 'first come, first served' as a basis for determining access to scarce, life-sustaining medical resources |
| Harris County Public Health and Environmental Services[48] | Report on the views of citizens from Harris Country Texas on distributing scarce healthcare resources (vaccines, anti-virals and ventilators) during a pandemic | Priority based on likelihood of recovery (occupation and age were given the lowest level of importance) |
| Silva et al[49] | Three public town hall meetings across Canada exploring perspectives on priority setting during an influenza pandemic | Life expectancy and socioeconomic status *should not* be considerations in allocating ventilators during an influenza pandemic |
| Diederich et al[35] | Germany-based survey on age as a criterion for setting priorities in healthcare | Found little evidence that age is accepted by the German public as a criterion relevant to prioritising healthcare |
| Daugherty Biddison et al[19] | Maryland-based pilot study in 2012 | Those most likely to survive and those who are valuable to others in a pandemic |
| Krütli et al[50] | Switzerland-based survey conducted between Dec 2013 and May 2014 using hypothetical situations of scarcity regarding (1) donor organs, (2) hospital beds during an epidemic and (3) joint replacements | 'Sickest first' was prioritised. 'Lottery', 'reciprocity', 'instrumental value' and 'monetary contribution' were considered very unfair allocation principles |
| Biddison et al[20] | Conducted in 2012 and 2014, Maryland residents' views on allocating scarce ventilators during an influenza pandemic | Priority based on short-term and long-term survival, though not exclusively; concerns raised about withdrawal of a ventilator in order to benefit another patient |
| Schoch-Spana et al[33] | Texas-based public engagement initiative | Those most likely to survive the current illness and those who will live longer, with emphasis on parents with dependents, and children |
| Huang et al[36] | US-based, conducted in 2020 using veil-of-ignorance reasoning in COVID-19 ventilator dilemmas | Prioritised younger over older patients with COVID-19 |
| Buckwalter and Peterson[34] | US-based survey conducted in 2020 investigating public attitudes toward allocating scarce resources during the COVID-19 pandemic | Priority-based survival chance and on seriousness of condition, but not when these entail reallocation between existing patients, or when they disadvantage at risk groups |

a better prognosis. Although there was somewhat higher support for prioritisation involving *withholding*, a clear majority of our respondents were prepared to withdraw treatment from a patient who had been receiving treatment in intensive care, but who had a significantly lower chance of survival than another patient currently needing treatment. Eighty-six per cent of our respondents supported this where there was a very large difference between patients in chance of survival, even though withdrawal of treatment would lead to death. This appears to be a strong endorsement of a utilitarian approach to management of intensive care beds in a pandemic. It is

somewhat in contrast to Buckwalter and Peterson, who found relative ambivalence for triage policies that would reallocate treatment in order to save more lives.[34] In the Maryland and Texas community studies, a number of participants expressed concern about withdrawal of a ventilator in order to benefit another patient. Nevertheless, 62% of participants in both studies accepted that there were circumstances where this was acceptable.[20 33] There may be relevant differences in community attitudes to treatment decisions or healthcare. UK respondents may be more familiar with the need for resource allocation in a publicly funded healthcare system and less averse to withdrawing treatment than those in the USA. Differences may also relate to the distinction between expressing reluctance or disquiet about a general policy, and a forced choice scenario, where participants were required to make decisions about which patient would survive.

## Limitations

As with any online survey, there are challenges in generalising to the wider community. In this case, while those who participated were part of pre-existing marketing research panels, they were representative of the UK general population for gender, age, income, education and employment.

The scenarios presented to participants in this survey are necessarily unrealistic. They were designed to assess the impact of single variables in the absence of uncertainty. This means that responses only indicate which factors participants would hypothetically take into consideration, but not how much relative weight respondents would give to different factors, or how they would decide where factors (eg, prognosis) were uncertain. In real cases, estimates of chance of survival, length of treatment or of life expectancy would necessarily include a margin of error, and these estimates may overlap between patients presenting for treatment. We described degrees of pre-existing disability and frailty in terms of their functional effect (since this features in standard severity definitions). However, this makes it difficult to assess whether participants were responding to the underlying condition (eg, frailty) or the resulting functional dependency.

We had added a control scenario where participants were asked to allocate treatment between patients from different racial backgrounds (who were specified to have a similar chance of survival). This indicated (as anticipated) that a majority of respondents would give each patient an equal chance of receiving treatment. However, as the pandemic unfolded, reports of racial disparities in infection and mortality rates increased.[37] This may have led some respondents to prioritise those of Black Caribbean ancestry, since they appear to be left worse off by the pandemic or to deprioritise such patients because of a belief (contrary to the details provided) that their outcome would be worse.

## Interpretation

The results of this survey suggest that members of the UK general public would support a broadly utilitarian approach to triage in the face of overwhelming need.[17 38] A majority elected to treat patients in ways that would maximise the outcome of intensive care treatment—in terms of numbers of lives saved, but also in terms of numbers of life-years saved and quality-adjusted life-years. From behind a 'veil of ignorance' (a device designed to remove partiality), they strongly endorsed a policy of prioritising patients with higher chance of survival. A very small proportion of respondents, if forced to choose between patients, elected to toss a coin to decide, potentially endorsing an egalitarian approach. This proportion increased where the difference between patients was relatively small. A tiny minority selected to treat a patient with worse predicted outlook. In scenarios relating to frailty, age or disability, it is possible that such responses reflected prioritarian concern for the worse off.

Our survey suggests that the UK public would potentially endorse the relevance of some patient factors for triage that are currently recommended in professional guidance documents relating to COVID-19 triage. For example, NICE guidance recommended the use of *clinical frailty* in decision-making.[8] British Medical Association (BMA) guidance referred to the importance of *chance of survival* as well as expected *duration of treatment*.[39] Intensive Care Society guidance recommended that in the event of extreme resource shortage, *patient age, comorbidities* and *frailty* might be relevant to assessment of the capacity of the patient to benefit (ie, survive).[40]

However, the members of the public surveyed in this study also clearly indicated the relevance for decisions of factors that do not appear in current guidance. That includes expected *duration of survival*, *patient age* (independent of chance of survival), *disability* (at least if severe/profound), *healthcare worker status* and *dependents*.

Our survey respondents expressed clear support for the permissibility of withdrawing treatment from a patient who already received a period of treatment in intensive care in favour of another patient with higher chance of survival. That is highly relevant to some of the ethical debates that have taken place during the pandemic. While BMA guidance expressed in-principle support for withdrawal of treatment in order to treat other patients, this has been criticised by a number of authors.[41–43]

Of course, surveys of the public's views do not settle ethical questions about what triage policy should be adopted. The public might misunderstand the relevant factual or ethical considerations, or there may be strong ethical arguments against inclusion of some factor that the public supports. However, the views of the wider community are relevant to ethical deliberation in a number of ways. Where ethical arguments and the views of the public converge in suggesting support for a factor in triage, that suggests that it should be strongly considered. For example, the general public's preferences in our survey would be consistent with arguments

and proposed ventilator allocation algorithms that aim to maximise healthcare benefit in the setting of overwhelming demand.[17][38] The results of this study suggest that if they wish to align with the values of the general public, professional bodies should consider including additional factors in UK pandemic triage guidance and more strongly endorse the permissibility of withdrawal and reallocation of treatment.[44] There will be a need to consider the potential for moral distress, should professionals be required to take steps like this that depart from usual ethical standards.[9]

Fortunately, while the UK had the highest number of deaths in Europe in the first wave of the pandemic, ICUs were not overwhelmed and it did not prove necessary to invoke specific triage protocols. However, there remains significant concern about subsequent waves of the virus in the coming months, particularly over winter, and there may yet need to be difficult decisions about which patients to provide with scarce treatment. Furthermore, the basic ethical principles relating to triage decisions are relevant for intensive care even outside the setting of a pandemic.

Although our study provides insights into which factors the public consider potentially relevant to triage decisions, it does not provide direct insight into the relative weight of those different factors. In reality, patients presenting in need of intensive care admission, even if they present simultaneously, will vary in a range of competing and overlapping ways. There is a need for guidance to help clinicians decide between patients who may have better outlook in some ways and worse in others. It will be helpful to further assess how the public balances patient factors when they compete in more complex triage decisions.

**Author affiliations**
[1]Oxford Uehiro Centre for Practical Ethics, Faculty of Philosophy, University of Oxford, Oxford, UK
[2]Newborn Care Unit, John Radcliffe Hospital, Oxford, UK
[3]Wellcome Centre for Ethics and Humanities, University of Oxford, Oxford, UK
[4]Murdoch Children's Research Institute, Melbourne, Victoria, Australia
[5]Department of Psychology, School of Arts and Social Sciences, City University of London, London, UK
[6]Kenan Institute for Ethics, Department of Philosophy, Duke University, Durham, North Carolina, USA
[7]Faculty of Law, University of Melbourne, Melbourne, Victoria, Australia

**Acknowledgements** The authors acknowledge very useful feedback from the MADLAB at The Kenan Institute for Ethics, Duke University.

**Contributors** DW—lead author responsible for the original idea for the study, playing a central role in formulating the survey questions, analysing the results and drafting the manuscript. HZ—contributed significantly to the formulation of survey questions, the preparation of the survey, analysis of results and the drafting of the manuscript. AK—significant contribution to the formulation of survey questions and to carrying out all associated statistical analyses. WS-A—significant contribution to the formulation of the survey and methodology, helping with the drafting of all sections. JS—significant contribution to the formulation of the survey and methodology, helping with the drafting of all sections.

**Funding** This study was supported by a grant from the University of Oxford Medical Sciences Division COVID-19 Research Response Fund #0008940. DW and JS were supported for this work by a grant from the Wellcome trust 203132/Z/16/Z. JS was supported by Wellcome trust grant 104848/Z/14/Z and, through his involvement with the Murdoch Children's Research Institute, was supported by the Victorian Government's Operational Infrastructure Support Program.

**Competing interests** None declared.

**Patient consent for publication** Not required.

**Ethics approval** The experiment was approved by the University of Oxford Central University Research Ethics Committee (R69537/RE001).

**Provenance and peer review** Not commissioned; externally peer reviewed.

**Data availability statement** Data are available in a public, open access repository. All data and materials for the survey are available through the Open Science Framework https://osf.io/gta3k/.

**ORCID iD**
Dominic Wilkinson http://orcid.org/0000-0003-3958-8633

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
