## [Reviewer comments · BMJ Open]

ARTICLE DETAILS

TITLE (PROVISIONAL)	Which factors should be included in triage? An online survey of the attitudes of the UK general public to pandemic triage dilemmas.
AUTHORS	Wilkinson, Dominic; Zohny, Hazem; Kappes, Andreas; Sinnott-Armstrong, Walter; Savulescu, Julian

VERSION 1 – REVIEW

REVIEWER	René Robert Intensive Care Unit University hospital, Poitiers, France
REVIEW RETURNED	15-Oct-2020

GENERAL COMMENTS	The study presented by Wilkinson and co-authors aims to evaluate the public opinion faced to triage choices that could be made by intensivists during pandemia. The strength of such a study is to give the opportunity to general public to express their opinion related to triage choices in this particular situation of pandemia and scarcity of ICU beds. I have several general concerns 1- it is not explained to the respondents that the chosen key variables are surrounded with an important risk of error chance at the individual level: e.g. of survival, life expectancy. Similarly the expected length of treatment cannot be adequately prognosticated. Lack of sensitivity or specificity of the scores that can be used to evaluate the prognosis at the individual level has been repeatedly underlined. Age is also becomes a potentially easy operational cursor, which we do not know how to place rationally. Long term life expectancy prediction cannot be proposed as a criterion without giving the error standard deviation of such prediction (fig 3) In other words, the real tools and their significance are not really given to public through the different choices: there is no clear data allowing to precisely asses the chance of survival for a patient. 2- the ethical dilemma analysis that should be more deeply part of the discussion section, underlying that in several selected situations, the choice is given between bad options. Including that strategies must assume “mistake of prophecy” and the eventual sacrifice of wrongly-predicted patients. (see Robert R et al Ann Intensive Care 2020) 3- the balance of the chosen scenario mixing some scenario with apparent “easy choices” and more complicated scenario. Finally at the bed-side easy scenarios most often lead to easy choices and more complex scenario cannot be solved without a multidimensional approach (with all the items selected by the authors) 4 other comments
--

	-Page 10 and Figure 5: the first scenario (1 day vs 5 days) appears to be inaccurate because one can assume that in such situation the predictability of needing only one day of ICU with mechanical ventilation is very low. Thus this is a major bias for this specific item -the alternative way to “cursor strategies” was “toss a coin”, in other studies the choice was first come, first served please comment
--	---

REVIEWER	Andrew Peterson George Mason University, United States
REVIEW RETURNED	24-Oct-2020

GENERAL COMMENTS	This is a well-written and well-designed study on public attitudes toward triage dilemmas in the Covid-19 pandemic. The study will be of interest to the readers of BMJ Open and may provide critical insight into public attitudes as Covid-19 cases increase during the winter months. Importantly, the study builds upon other recent survey studies regarding triage dilemmas and makes several notable improvements to those survey designs, namely in requiring participants to make forced choices between patients based on particular factors. Below, I outline several comments that may assist the authors in improving the manuscript.  1. If space is possible, the article summary section might be improved by including an additional limitation bullet point such as: “survey findings do not allow assessment of all intersections of different patient factors” The survey appears to treat patient factors as independent. But allocation decisions may ultimately be made based on the interaction of these patient factors. 2. Page 5: Use of the word “utilitarian”. I agree with this characterization of the maximizing benefits principle. But it is worth acknowledging that some experts disagree with this characterization. These experts argue that non-utilitarian considerations might also justify maximizing benefits in terms of lives, or life-years, saved. See, for example: Wasserman D, Persad G, Millum J. Setting priorities fairly in response to Covid-19: identifying overlapping consensus and reasonable disagreement. Journal of Law and the Biosciences. 2020 Jan;7(1):Isaa044. The authors acknowledge this to some degree in footnote 1. 3. Page 5: A general comment of caution regarding framing of “quality of life” and “quality adjusted life years” in the context of triage. The authors write: “For example, prioritising patients with a longer life expectancy or less pre-existing disability would not save more lives, but would result in more quality-adjusted life years.[16]” Two issues seem to be run together here: preservation of more life years and preservation of more QALYs. The US triage literature has focused on prioritizing short-term life expectancy post discharge, not whether a patient accrues greater wellbeing or is expected to have a greater quality of life after treatment. The worry is that quality of life judgments in triage could be ad hoc and result in biases against people with pre-existing disabilities. Discussion of quality of life judgments comes up in several other places throughout the manuscript and survey design. Briefly making these issues explicit would be helpful for readers who are not entirely familiar with the triage literature.
--

4. Page 13: The authors write, regarding the Buckwalter and Peterson 2020 survey, “The ‘prioritarian’ policy was described as directing therapy to those most seriously ill, but it was unclear whether respondents understood that such a policy would potentially save fewer lives (since sicker patients often have a lower chance of survival) or intuitively believed that sicker patients would benefit most from treatment.” As a coauthor of the Buckwalter and Peterson study, I agree with this criticism that our design might not have sufficiently distinguished ordinary judgments related to maximizing benefits versus prioritizing patients who are the worst off. As the authors suggest, this could lead to participants favoring a “prioritarian” policy even though their underlying psychology might be utilitarian oriented. Nonetheless, contrary to the authors’ description of the study, the relevant survey item in Buckwalter and Peterson 2020 related to “prioritarian” policy does suggest that fewer lives will be saved when resources are scarce. The survey item states (available in the supplemental materials and on OSF): “The triage team at County General Hospital is responsible for deciding the order that new patients receive lifesaving resources, such as ventilators or ICU beds. The team has recently instituted a new policy. According to this policy, patients will receive lifesaving resources in the order of the seriousness of illness, with those who are the worst off being prioritized. County General is not well supplied, so there are many more patients than there are lifesaving resources available. Because of this, many patients will go without lifesaving treatment who need them.” Additionally, based on comments from reviewers, the term “prioritarian” was removed from the Buckwalter and Peterson manuscript and replaced with “prioritizing the worst off,” consistent with the language presented to survey participants. Very minor revisions to this section, which still highlight the relevant criticism of Buckwalter and Peterson, may be helpful for readers.

5. General comment on framing of participant responses to disability and frailty conditions. The disability condition in the survey materials provided in the BMJ Open submission describes the patient as having “...profound and multiple learning disabilities. This patient has difficulty seeing, hearing, and moving and is dependent on others for self-care. This patient lives in a nursing home and has the mental age of less than 3 years old.” The survey materials available on the Open Science Framework for one of the frailty conditions describe “two older patients who have existing health problems that mean they are frail and both require ICU admission for coronavirus. The patients are otherwise similar in features. Patient Q is severely frail and is aged 66. He or she is completely dependent on others for personal care. Patient R is mildly frail and is aged 82. This means that they have some evident slowing and need help with some higher order activities of daily living (finances, transportation, heavy housework, medications).” I reference both of these items as examples of the overall disability and frailty conditions.

I have a general concern about the formulation of these conditions and how subsequent participant responses are presented in the manuscript. These survey items seem to run together frailty or disability with dependency. Frailty and disability are, of course, associated with dependency, but they do not imply dependency. So, based on the survey question, how do we know whether

	participants were responding to the patient’s described disability or frailty, on the one hand, or their degree of dependency, on the other, or both? It may be that participants were more inclined to provide less priority to patients who are described as being more dependent of social support services, but not because they have disabilities or are frail. I think it’s important to make this distinction explicit. As the manuscript currently reads, it seems to suggest that respondents favored giving less priority to people with disabilities (using disability as an example) because of their disability, or perceived diminished quality of life, not because of their dependency on social support services. But it is plausible that respondents were, instead, reacting to these varying degrees of dependency and how that stresses social support services, particularly in the UK. It’s worth asking whether participants would have responded the same way if issues of dependency were not emphasized as much in the survey items. The authors may have accounted for this. But, as far as I can tell, it doesn’t come through in the manuscript. A brief and nuanced acknowledgement of this in the discussion and results section would improve the manuscript. The authors should also check that all figures for the manuscript are included and labeled correctly. When examining the above issue, I looked for figure 6 for the disability condition results, referenced on page 10. But this doesn’t appear to be included in the BMJ Open submission. 6.The last study listed in the table, Buckwalter and Peterson (2020), is forthcoming at Plos One. The doi assigned by the journal is: 10.1371/journal.pone.0240551.
--	--

VERSION 1 – AUTHOR RESPONSE

Reviewer 1

1.1- it is not explained to the respondents that the chosen key variables are surrounded with an important risk of error chance at the individual level: e.g. of survival, life expectancy. Similarly the expected length of treatment cannot be adequately prognosticated. Lack of sensitivity or specificity of the scores that can be used to evaluate the prognosis at the individual level has been repeatedly underlined.

Age is also becomes a potentially easy operational cursor, which we do not know how to place rationally.

Long term life expectancy prediction cannot be proposed as a criterion without giving the error standard deviation of such prediction (fig 3)

In other words, the real tools and their significance are not really given to public through the different choices: there is no clear data allowing to precisely asses the chance of survival for a patient.

RESPONSE:

Thank-you for this helpful comment. Because we were seeking the views of non-experts, and aiming to evaluate the relevance of different patient factors in a controlled way (a philosophical thought experiment) we had to present simplified and somewhat unrealistic cases in the survey.

We have added to the methods this clarification:

“The patients differed in one of six key variables: chance of survival, life expectancy, age, expected length of treatment, disability and degree of frailty (see Figure (1)). For a given scenario, variables were assumed to be known or predicted accurately and (unless specified) other characteristics were the same between patients.”

We had previously noted the unrealistic nature of cases in the study limitations, however, we have expanded this comment to acknowledge the insights of the reviewer.

“The scenarios presented to participants in this survey are necessarily unrealistic. They were designed to assess the impact of single variables in the absence of uncertainty. This means that responses only indicate which factors participants would hypothetically take into consideration, but not how much relative weight respondents would give to different factors, or how they would decide where factors (eg prognosis) were uncertain. In real cases, estimates of chance of survival, length of treatment or of life expectancy would necessarily include a margin of error, and these estimates may overlap between patients presenting for treatment.”

1.2- the ethical dilemma analysis that should be more deeply part of the discussion section, underlying that in several selected situations, the choice is given between bad options. Including that strategies must assume “mistake of prophecy” and the eventual sacrifice of wrongly-predicted patients. (see Robert R et al Ann Intensive Care 2020)

RESPONSE:

Thank-you for highlighting this very useful reference, including valuable reflections on the profound challenges faced by front line clinicians in the first wave of the pandemic. We did not mean to diminish the profound personal challenges that clinicians faced in conducting triage during the pandemic. We have added to the manuscript to draw attention to the challenges that clinicians have faced already and would face if guidelines were changed (eg to support reallocation of treatment.

Introduction

“Faced with such concern, health systems around the world prepared guidance for intensive care triage.[6–8] The aim was to help health professionals make extremely difficult and potentially distressing[9] decisions about which patients to admit to ICU and treat with mechanical ventilation.”

Implications

“The results of this study suggest that if they wish to align with the values of the general public, professional bodies should consider including additional factors in UK pandemic triage guidance and more strongly endorse the permissibility of withdrawal and reallocation of treatment.[47] There will be a need to consider the potential for moral distress, should professionals be required to take steps like this that depart from usual ethical standards[9]

1.3 the balance of the chosen scenario mixing some scenario with apparent “easy choices” and more complicated scenario.

Finally at the bed-side easy scenarios most often lead to easy choices and more complex scenario cannot be solved without a multidimensional approach (with all the items selected by the authors)

RESPONSE:

Thank you for this insightful comment. We agree that the scenarios as presented were unrealistic and much easier than complex real cases. We have highlighted this in our limitations section, as well as the need for further studies exploring response to more complicated scenarios with multiple variables.

“The scenarios presented to participants in this survey are necessarily unrealistic. They were designed to assess the impact of single variables in the absence of uncertainty. This means that responses only indicate which factors participants would hypothetically take into consideration, but not how much relative weight respondents would give to different factors, or how they would decide where factors (eg prognosis) were uncertain. In real cases, estimates of chance of survival, length of treatment or of life expectancy would necessarily include a margin of error, and these estimates may overlap between patients presenting for treatment.”

Conclusion

“There is a need for guidance to help clinicians decide between patients who may have better outlook in some ways and worse in others. It will be helpful to further assess how the public balances patient factors when they compete in more complex triage decisions.

1.4 other comments

-Page 10 and Figure 5: the first scenario (1 day vs 5 days) appears to be inaccurate because one can assume that in such situation the predictability of needing only one day of ICU with mechanical ventilation is very low. Thus this is a major bias for this specific item

RESPONSE:

Thank-you for this. We appreciate in retrospect that this scenario was unrealistic in relation to a patient with COVID needing respiratory support. (Though other patients potentially competing for an intensive care bed might be predicted to need only a short duration of support – eg a patient following a major surgical procedure).

As there was no significant difference between the response to this scenario, and other (potentially more realistic) versions of the scenario (1week vs 2 weeks, 1 week vs 10 weeks), it does not appear to have influenced participants’ response.

We have noted in the limitations section that some of the scenarios were unrealistic.

“The scenarios presented to participants in this survey are necessarily unrealistic”

1.5 -the alternative way to “cursor strategies” was “toss a coin”, in other studies the choice was first come, first served please comment

RESPONSE

Thank you for this question. We chose to give respondents the option of “tossing a coin” as we had used this in our own previously published survey on resource allocation decisions in critical care.

“It was adapted from a previous survey on rationing in neonatal intensive care[25].”

The advantage of “tossing a coin” is that this most closely resembles a truly egalitarian approach to allocation of treatment. We appreciate that in practice clinicians may resort to “First come, first served” if they do not choose between patients. However, “First come, first served” gives priority to those who happen to arrive earlier and may therefore advantage those with greater access to medical care.(for example, see reference to Wasserman in the revised manuscript).

Reviewer: 2

2.1 This is a well-written and well-designed study on public attitudes toward triage dilemmas in the Covid-19 pandemic. The study will be of interest to the readers of BMJ Open and may provide critical insight into public attitudes as Covid-19 cases increase during the winter months. Importantly, the study builds upon other recent survey studies regarding triage dilemmas and makes several notable improvements to those survey designs, namely in requiring participants to make forced choices between patients based on particular factors. Below, I outline several comments that may assist the authors in improving the manuscript.

RESPONSE: Many thanks for these very positive and encouraging comments.

2.2 If space is possible, the article summary section might be improved by including an additional limitation bullet point such as: “survey findings do not allow assessment of all intersections of different patient factors” The survey appears to treat patient factors as independent. But allocation decisions may ultimately be made based on the interaction of these patient factors.

RESPONSE:

Thank you for this suggestion. We have added to the Article summary as you suggest.

- “Survey findings do not allow assessment of relative weight of different factors or how they might interact”

2.3 Page 5: Use of the word “utilitarian”. I agree with this characterization of the maximizing benefits principle. But it is worth acknowledging that some experts disagree with this characterization. These experts argue that non-utilitarian considerations might also justify maximizing benefits in terms of lives, or life-years, saved. See, for example: Wasserman D, Persad G, Millum J. Setting priorities fairly in response to Covid-19: identifying overlapping consensus and reasonable disagreement. *Journal of Law and the Biosciences*. 2020 Jan;7(1):lsaa044. The authors acknowledge this to some degree in footnote 1.

RESPONSE

Thank you for pointing us to David Wasserman’s excellent paper.

We have added reference to it and modified this section of the manuscript.

“Saving as many lives as possible is arguably a fundamental ethical principle for any triage framework, and supported by overlapping consensus of different ethical theories.[15][16]”

Footnote 1: “... Prioritarianism might imply priority for patients who are worse off in other ways (for example having experienced social or economic disadvantage). Non-utilitarian theories may also support maximizing numbers of lives or life-years saved. [15]”

2.4 Page 5: A general comment of caution regarding framing of “quality of life” and “quality adjusted life years” in the context of triage. The authors write: “For example, prioritising patients with a longer life expectancy or less pre-existing disability would not save more lives, but would result in more quality-adjusted life years.[16]” Two issues seem to be run together here: preservation of more life years and preservation of more QALYs. The US triage literature has focused on prioritizing short-term life expectancy post discharge, not whether a patient accrues greater wellbeing or is expected to have a greater quality of life after treatment. The worry is that quality of life judgments in triage could be ad hoc and result in biases against people with pre-existing disabilities. Discussion of quality of life judgments comes up in several other places throughout the manuscript and survey design. Briefly making these issues explicit would be helpful for readers who are not entirely familiar with the triage literature.

RESPONSE:

Thank-you for highlighting this gap in the discussion. We have added to the introduction to note this concern.

“For example, prioritising patients with a longer life expectancy or less pre-existing disability would not save more lives, but would result in more quality-adjusted life years.[16] However, inclusion of such factors might be vulnerable to bias, and raise concerns about discrimination.[15]”

We have also added a comment to the discussion related to this

“In the current survey, respondents appeared more inclined to toss a coin when choosing between patients with different types or degrees of disabilities (~40% of respondents in three scenarios in this survey, compared to ~20% in the previous survey). This might reflect public concern about the ethical problems in assessing quality of life,[15] or a desire to avoid discrimination.”

2.5 Page 13: The authors write, regarding the Buckwalter and Peterson 2020 survey, “The ‘prioritarian’ policy was described as directing therapy to those most seriously ill, but it was unclear whether respondents understood that such a policy would potentially save fewer lives (since sicker patients often have a lower chance of survival) or intuitively believed that sicker patients would benefit most from treatment.” As a coauthor of the Buckwalter and Peterson study, I agree with this criticism that our design might not have sufficiently distinguished ordinary judgments related to maximizing

benefits versus prioritizing patients who are the worst off. As the authors suggest, this could lead to participants favoring a “prioritarian” policy even though their underlying psychology might be utilitarian oriented. Nonetheless, contrary to the authors’ description of the study, the relevant survey item in Buckwalter and Peterson 2020 related to “prioritarian” policy does suggest that fewer lives will be saved when resources are scarce. The survey item states (available in the supplemental materials and on OSF): “The triage team at County General Hospital is responsible for deciding the order that new patients receive lifesaving resources, such as ventilators or ICU beds. The team has recently instituted a new policy. According to this policy, patients will receive lifesaving resources in the order of the seriousness of illness, with those who are the worst off being prioritized. County General is not well supplied, so there are many more patients than there are lifesaving resources available. Because of this, many patients will go without lifesaving treatment who need them.” Additionally, based on comments from reviewers, the term “prioritarian” was removed from the Buckwalter and Peterson manuscript and replaced with “prioritizing the worst off,” consistent with the language presented to survey participants. Very minor revisions to this section, which still highlight the relevant criticism of Buckwalter and Peterson, may be helpful for readers.

RESPONSE:

Many thanks for this helpful clarification and suggestion. We have modified this section of the manuscript.

“Participants indicated support for triage policies that aimed to save the most lives (“utilitarian” policy), or treat the sickest patients (labelled “prioritizing the worst off”), but disagreed with policies that treated patients in order of arrival (“egalitarian”), or prioritized based on social importance.[40] However, the results of Buckwalter’s study are hard to compare with our own. Participants in that study were asked to endorse general policies, but not presented with specific cases of competing patients. The policy descriptions might not have sufficiently distinguished ordinary judgments related to maximizing benefits versus prioritizing patients who are the worst off since some participants who chose the “prioritizing the worst off” policy might have intuitively believed that sicker patients would benefit most from treatment (though in fact sicker patients may have a lower chance of survival).”

2.6 General comment on framing of participant responses to disability and frailty conditions. The disability condition in the survey materials provided in the BMJ Open submission describes the patient as having “...profound and multiple learning disabilities. This patient has difficulty seeing, hearing, and moving and is dependent on others for self-care. This patient lives in a nursing home and has the mental age of less than 3 years old.” The survey materials available on the Open Science Framework for one of the frailty conditions describe “two older patients who have existing health problems that mean they are frail and both require ICU admission for coronavirus. The patients are otherwise similar in features. Patient Q is severely frail and is aged 66. He or she is completely dependent on others for personal care. Patient R is mildly frail and is aged 82. This means that they have some evident slowing and need help with some higher order activities of daily living (finances, transportation, heavy housework, medications).” I reference both of these items as examples of the overall disability and frailty conditions.

I have a general concern about the formulation of these conditions and how subsequent participant responses are presented in the manuscript. These survey items seem to run together frailty or disability with dependency. Frailty and disability are, of course, associated with dependency, but they do not imply dependency. So, based on the survey question, how do we know whether participants were responding to the patient’s described disability or frailty, on the one hand, or their degree of dependency, on the other, or both? It may be that participants were more inclined to provide less priority to patients who are described as being more dependent of social support services, but not because they have disabilities or are frail. I think it’s important to make this distinction explicit. As the manuscript currently reads, it seems to suggest that respondents favored giving less priority to people

with disabilities (using disability as an example) because of their disability, or perceived diminished quality of life, not because of their dependency on social support services. But it is plausible that respondents were, instead, reacting to these varying degrees of dependency and how that stresses social support services, particularly in the UK. It's worth asking whether participants would have responded the same way if issues of dependency were not emphasized as much in the survey items. The authors may have accounted for this. But, as far as I can tell, it doesn't come through in the manuscript. A brief and nuanced acknowledgement of this in the discussion and results section would improve the manuscript.

RESPONSE:

Thank-you for this very helpful and insightful suggestion. We had attempted to use standardised clinical definitions of severity of frailty and disability (<https://www.ncbi.nlm.nih.gov/books/NBK332877/>) but make them accessible to participants without a medical background. Clinical frailty, according to the scale recommended by the NHS is defined formally in terms of function and dependency (https://www.bgs.org.uk/sites/default/files/content/attachment/2018-07-05/rockwood_cfs.pdf). We have clarified this in the methods “An approximate equivalent ‘mental age’ was included alongside the description of function to aid participants’ understanding, though this no longer features in official classification of intellectual/learning disability.[30] Degrees of frailty were described in accordance with the Clinical Frailty Score.”

We accept that our descriptions of frailty and disability could have been improved (and plan to use different descriptors in future studies). We have added acknowledgement of this to our limitations section in the discussion.

“We described degrees of pre-existing disability and frailty in terms of their functional effect (since this features in standard severity definitions). However, this makes it difficult to assess whether participants’ were responding to the underlying condition (eg frailty), or the resulting functional dependency.”

2.7 The authors should also check that all figures for the manuscript are included and labeled correctly. When examining the above issue, I looked for figure 6 for the disability condition results, referenced on page 10. But this doesn't appear to be included in the BMJ Open submission.

RESPONSE:

Many thanks for drawing this to our attention. We have attached the figure to the revised manuscript and double checked the other figures and legends.

2.8 The last study listed in the table, Buckwalter and Peterson (2020), is forthcoming at Plos One. The doi assigned by the journal is: 10.1371/journal.pone.0240551.

RESPONSE:

Thank-you for letting us know. We have updated this reference.

VERSION 2 – REVIEW

REVIEWER	R Robert Department of Critical Care, University hospital of Poitiers France
REVIEW RETURNED	13-Nov-2020

GENERAL COMMENTS	author made appropriate answers to my comments.
REVIEWER	Andrew Peterson George Mason University
REVIEW RETURNED	11-Nov-2020
GENERAL COMMENTS	The authors have addressed the issues raised in my initial review. This is an important and well-designed study. I look forward to seeing it published.